# The Effect of the Season, the Maintenance System and the Addition of Polyunsaturated Fatty Acids on Selected Biological and Physicochemical Features of Rabbit Fur

**DOI:** 10.3390/ani12080971

**Published:** 2022-04-08

**Authors:** Katarzyna Roman, Martyna Wilk, Piotr Książek, Katarzyna Czyż, Adam Roman

**Affiliations:** 1Department of Animal Nutrition and Feed Science, Wrocław University of Environmental and Life Sciences, 25 C.K. Norwida St., 51-630 Wrocław, Poland; katarzyna.roman@upwr.edu.pl; 2Independent Researcher, 51-649 Wrocław, Poland; pioksiazek@student.agh.edu.pl; 3Division of Sheep and Fur Animals Breeding, Wrocław University of Environmental and Life Sciences, 25 C.K. Norwida St., 51-630 Wrocław, Poland; katarzyna.czyz@upwr.edu.pl; 4Department of Environment Hygiene and Animal Welfare, Wrocław University of Environmental and Life Sciences, 25 C.K. Norwida St., 51-630 Wrocław, Poland; adam.roman@upwr.edu.pl

**Keywords:** rabbit, ethyl esters, linseed oil, fatty acids, hair coat

## Abstract

**Simple Summary:**

Rabbit furs are a valuable material used in the fur industry. Many studies show beneficial effects of omega-3 acids supplementation on the skin and coat of animals. The aim of the study was to show the impact of environmental conditions and dietary supplementation with ethyl esters of linseed oil on the quality of the rabbit hair coat. The experiment was carried out in four stages: laboratory (summer and winter) and outdoor (summer and winter). The experimental rabbits were given an addition of ethyl linseed oil to their feed (during 2 months). To assess biological and physico-mechanical properties of the coat samples and to determine fatty acid profile and histological evaluation, the hair samples were collected three times: before the study, after two months of treatment, and after two months from the end of supplementation. The obtained results show that the environmental conditions have a major impact on the quality of the rabbit coat. The best results of hair heat protection were obtained from animals kept outdoors. Administration of linseed oil ethyl esters had a positive effect on the hair fatty acid profile.

**Abstract:**

The aim of the study was to show the impact of environmental conditions and dietary supplementation with ethyl esters of linseed oil on the quality of the rabbit hair coat. The research was divided into 4 stages: laboratory (summer and winter) and outdoor (summer and winter). In each stage of the research, animals were divided into control and experimental groups. The animals were fed in accordance with the feeding standards of reproductive rabbits during the period of sexual dormancy. The rabbits from the experimental groups during the first two months were given an addition of ethyl linseed oil to the feed. In the experiment, linseed oil was cold-pressed directly in the laboratory. Three samples of hair were collected: before the study, after two months of treatment, and after two months from the end of supplementation. The hair coat biological properties, such as share of individual hair fractions (%), heat transfer index (HTI), hair diameter (μm), as well as physico-mechanical properties such as breaking force (N), breaking stress (kg/cm^2^) and elongation (%) were performed. Moreover, the histological structure of hair and histological hair evaluation were performed. The fatty acid profile was determined in the hair as well. The obtained results of the content of individual fatty acids were grouped into saturated fatty acids and unsaturated fatty acids. In addition, omega-3 and omega-6 were distinguished from the group of unsaturated acids. The environmental conditions have a major impact on the quality of the rabbit coat. The best results of hair thickness and their heat protection were obtained from animals kept outdoors. The studies did not show an influence of the administered preparation on the quality of the rabbit coat. The hair became thinner, but more flexible and tear-resistant. Administration of linseed oil ethyl esters had significant, beneficial changes in the fatty acid profile in hair and hair sebum were observed. There was a significant increase in omega-3 acids, and a significant decrease in the ratio of omega-6 to omega-3 acids.

## 1. Introduction

Domestic rabbits (*Oryctolagus cuniculus f. domestica*), are the earliest domesticated fur animals in the world. The first attempts to domesticate European rabbits (*O. cuniculus*) consisted in keeping them in a semi-wild state, in specially separated and fenced areas (lat. *leporarium*) [1,2].

Two basic raw materials can be obtained from rabbits: meat—the so-called “white meat”, and leather—for the production of furs, jackets, collars, and leather haberdashery [3]. The success in rearing and breeding rabbits is decided by many factors such as fashion, climate, and current demand for furs during a given season. Rabbit furs are an extremely valuable raw material used in the fur industry. Modern methods of fur and leather processing allow for the achievement of very good products with high market value. Compared to artificial furs, natural furs are fully biodegradable in mere years, which does not add to the pollution of the natural environment [4,5].

Nutrition is one of the base elements of rearing and breeding rabbits, which enables achievement of set production standards and effects. Rabbits are herbivorous creatures that require green feeds, root crops, and an appropriate amount of full-value concentrated feeds in order to achieve healthy growth. The base nutrition element, aside from protein and carbohydrates, is fat, that supplies fatty acids into the body. Their main division is made on the basis of the number of double bonds and distinguishes two basic groups: saturated fatty acids (SFA) and unsaturated fatty acids (UFA), which are divided into monounsaturated fatty acids (MUFA) and polyunsaturated fatty acids (PUFAs). The group of PUFAs can be divided into omega-3 (n-3) acids, which include extremely valuable α-linolenic acid (ALA) with its derivatives EPA (eicosapentaenoic acid) and DHA (docosahexaenoic acid), and omega-6 (n-6) acids, which include linoleic acid (LA) and its derivatives. The most important role for the human and animal body is played by omega-3 acids [6,7]. Neither acids from the ALA family nor acids from the LA family are synthesized in the human body and many animals, hence why they should be supplied from the outside with food [8,9]. Among the products of plant origin, the main source of n-3 acids are nuts, sesame seeds [10], linseed (about 50% ALA), and vegetable oils, e.g., soybean or rapeseed [11]. 

Linseed is an extremely rich source of valuable fatty acids. In seeds of traditional varieties, more than 80% of the sum of all fatty acids are PUFA, among which the vast majority is α-linolenic acid (about 60%). Thanks to this, the ratio of n-6/n-3 fatty acids in them is about 0.3. Linseed in meal form is given to livestock to enrich the feed with ALA. Many studies indicate that supplementation of animals with omega-3 acids has a beneficial effect on the skin and coat. Studies done on dogs have shown that the addition of linseed oil and linseed had a beneficial effect on hair growth rate and coat [12,13,14]. Small amounts of linseed are also added to feed for companion animals, e.g., songbirds and dogs, to improve the quality of plumage or coat [15]. Unfortunately, linseed also contains anti-nutrients such as linamarin and the enzyme linase that hydrolyzes it. However, the amount of linase and linmarine in linseeds can be reduced or completely removed through appropriate chemical processes, including the esterification process [15]. 

In this study, rabbits were used as model animals, in terms of research on the effect of omega-3 fatty acids on the state of the coat of fur animals. The aim of the study was to determine the effect of supplementation of the feed ration of rabbits with ethyl esters of linseed oil on selected features of the coat, including the profile of fatty acids. In addition, the conducted research was aimed at demonstrating the influence of changing environmental conditions, i.e., season and maintenance conditions, on selected physicochemical and biological parameters of the hair coat of termond white rabbits.

## 2. Materials and Methods

### 2.1. Animals

The experiment was conducted at the Wrocław University of Environmental and Life Sciences (Poland) and was divided into four stages (I, II, III, IV). Samples taken for the study consisted of hair coat samples obtained from rabbits of the termond breed kept in different systems during the summer (S) and winter (W), supplemented with the addition of ethyl esters of linseed oil. The tests were carried out in laboratory conditions (temperature approx. 18 °C, humidity approx. 65–70%), in single metal cages divided into boxes (I and II) and in production conditions, in external free-standing, two-story wooden cages (III and IV), equipped with a feeder and a droplet drinker. The cages met all animal welfare requirements and legal standards for keeping livestock [16]. 

For the experiments, males of the termond rabbit breed (about 3–4 months old) were used. Before the start of the study, all rabbits were examined by a veterinarian, dewormed, and then vaccinated against viral hemorrhagic rabbit disease and myxomatosis. After the adaptation period (2 weeks), the first samples were taken and ethyl ester supplementation began. Each stage of the experiment lasted 16 weeks: I-L-S (from June to September), II-L-W (from November to February), III-O-S (from June to September) and IV-O-W (from November to February). Feed administration continued for the first 8 weeks of each stage of the study. Hair coat samples were taken from the animals three times in each of the stages of the experiment: before the start of the study, 8 weeks after administration of the preparation, after another 8 weeks from the end of supplementation. After the end of the experiment, all rabbits were given for adoption to a private breeder.

### 2.2. Feeding

The animals were fed with a complete mixture, granulated (approx. 150 g of feed/day). The feed granules included: wheat bran, grass mixture, dried molasses beet pulp, sunflower post-extraction meal, rapeseed post-extraction meal, corn, alfalfa, beet molasses, post-extraction soybean meal (toasted), mineral-vitamin supplement. The percentage of nutrients in the feed was determined. Granules ingredients and chemical composition of granules shown Table 1.

All animals were provided with constant access to fresh water, hay and dried twigs from fruit trees (mainly apple and pear trees) [17]. The animals were fed in accordance with the feeding standards of reproductive rabbits during the period of sexual dormancy [18]. 

In the experiment, ethyl esters of polyunsaturated fatty acids obtained from linseed oil were used [19]. Linseed oil was cold-pressed, directly in the laboratory. A new batch of the test preparation was synthesized every 3 weeks. The preparation was stored in dark glass bottles in the refrigerator at 4 °C. Before starting supplementation with linseed oil ethyl esters, the addition of the preparation was tested on 10 rabbits not covered by experience for a period of 14 days. No disturbing symptoms were observed, e.g., loose stools, and the animals willingly ate feed with the preparation. 

In the administered granulate, hay and ethyl esters of linseed oil, the fatty acid profile was determined (Table 2).

### 2.3. Arrangement of Experience

In all stages of the experiment, 16 rabbits were used, which were randomly divided into two groups: control (C) and experimental (E), 8 in each. Group C received granulated feed without additives, group E received an additional 5 mL of linseed oil ethyl esters per each animal for the first 8 weeks of the study, esters were administered in the morning, directly to slightly crushed granulated feed. All feed along with the dose of the preparation was quickly eaten, no leftovers were found. For the next 8 weeks, all animals were given feed without additives. The dose of the supplement was determined to achieve a tenfold reduction of the ratio of omega-6 to omega-3 EFAs in the rabbits’ diet. The ratio of the above groups of acids after the addition of esters was about 1:1 (Table 3). This ratio of n-6:n-3 acids is considered the most favorable for the conversion of EPA and DHA from α-linolenic acid [20].

### 2.4. Physio-Mechanical Analysis of the Coat

The study evaluated the coat in terms of the share of individual hair fractions (%), heat transfer index (HTI), hair diameter (μm), physio-mechanical properties such as: breaking force (N), breaking stress (kg/cm^2^) and elongation (%), histological structure of hair, fatty acid profile (%). The samples taken for determination of hair participation in individual fractions and heat protection were hair cast from an area of 25 cm^2^ (5 cm × 5 cm) on the left side of the animal. Hair coat samples were taken once during each stage of the study, at the beginning of the experiment. The criterion for dividing hair into individual fractions was thickness, length and appearance. The above parameters were assessed using an illuminated laboratory lamp with a magnifier with a magnification of 20×. The amount of hair in individual fractions (%) was determined by separating the hair into down and ground cover hair using tweezers counting up to 1000 each sample. HTI was determined in two repetitions (about 2 g per each). The following formula was used to calculate the HTI:HTI = HSD/HSDo = (M × Cp × R/A × ∝)/HSDo(1)
where: HSD—density of heat flux falling on the test sample (KW/m^2^); HSDo—density of heat flux falling directly on the calorimeter (KW/m^2^); M—the period of calorimetry (kg); Cp—specific heat of aluminum 900 (J/kg °C); R—the rate of increase in the temperature of the calorimeter in the linear part of the graph (°C/s); A—calorimeter area (m^2^); ∝—absorption coefficient of the blackened surface of the calorimeter.

In order to assess the effect of ethyl esters on the physico-mechanical parameters of the coat, the samples were taken three times: on the day of commencement of the study, after 8 weeks of administration of the preparation, two months after the end of supplementation. For measurements, hair combed out of the back of animals was used. Subsequently, measurements of thickness, elongation, breaking force, and breaking stress were made. The diameter measurements were made with an MP3 lamanester at a magnification of 500×, assessing 100 down hair and 100 ground cover hair from each sample. 

The measurement of the breaking force needed to calculate the strength and elongation of the hair was performed on 30 randomly selected hairs, using the Matest electronic ripper and the computer program “Matest”. 

Measurements of the diameter and breaking force of the cover hair taken from rabbits during the research allowed us to calculate the value of the hair-breaking stress. This parameter is expressed as the ratio of the breaking force to the cross-sectional area of the hair.
N = P × 10^4^/π × d^2^ × 9.81(2)
where: N—breaking stress (kg/mm^2^); P—breaking force (cN); d—diameter of the hair section (μm).

### 2.5. Histological Analysis of Hair

To assess the effect of the preparation administered during the research, photographs of hair taken immediately before the start of supplementation and after a two-month period of supplementation were taken. Hair obtained from rabbits from experimental groups was used for the analysis. Cover and down hair were evaluated. Histological evaluation was made using the LEO 435VO Zeiss scanning microscope (Carl Zeiss SMTAG). Based on the photos taken, the characteristics of the cuticle layer and their cross-section in down and cover hair were made. The slides for the photos were cleaned with ether and alcohol and rinsed in a sound scrubber for about 5 min, and after drying sprayed with gold. On the basis of the SEM (scanning electron microscope) images of down and cover hair at ×400, ×1000, ×2000 and ×3000 magnification, the arrangement of scales in relation to the longitudinal axis of the hair, the type of cuticle, the structure of the edges of the cuticles, the distance between the edges of the cuticles and the structure of the edges of the cuticles were determined. 

### 2.6. Fatty Acid Profile

To investigate the effect of the administered preparation on the quality of rabbits’ coats, the profile of fatty acids contained in sebum covering the hair was analyzed (7890A, Agilent Technologies, Santa Clara, CA, USA). The obtained results of the content of individual fatty acids were grouped into saturated acids (SFA) and unsaturated acids (UFA, including monounsaturated PUFA and polyunsaturated MUFA). In addition, two subgroups were distinguished from the group of unsaturated acids—omega-3 (n-3) and omega-6 (n-6).

Hair coat samples for the analysis of the fatty acid profile were taken three times: on the day of commencement of the study, after 2 months of administration of the preparation, and after two months from the end of supplementation. For measurements, hair combed out of the back of animals was used. Fat from rabbits’ coats was extracted with ether by the Soxhlet method. Methyl esters of fatty acids were obtained according to the Christopherson–Glass methodology [21]. The fatty acid profile in the obtained samples was determined using a gas chromatograph (7890A, Agilent Technologies, Santa Clara, CA, USA) with an FID detector. The identification of the obtained fatty acids was carried out by comparison with the retention times of the standards of methyl esters of Supelco 37 fatty acids (Sigma Aldrich, Santa Clara, CA, USA).

### 2.7. Statistical Analysis

For each of the analyzed factors: additive—addition of linseed oil ethyl esters (C—control or E—experimental), condition—animal living conditions (L—laboratory or O—outdoor cage), and season—season of experiment (S—summer or W—winter), the tables present average values and standard deviation. The obtained data for main effects (additive, conditions, season) were analyzed by analysis of variance ANOVA using Statistica 13.3 (TIBCO Software Inc., Palo Alto, CA, USA). Significant differences between the groups were confirmed by Duncan’s multiple range test. Highly significant differences at the level of *p* < 0.01 were marked uppercase—A, B and significant differences at the level of *p* < 0.05 were marked lowercase—a, b.

The obtained data for main effects were analyzed by analysis of variance ANOVA using Statistica 13.3 (TIBCO Software Inc., Palo Alto, CA, USA). Significant differences between the groups were confirmed by Duncan’s multiple range test. Differences with *p* < 0.05 were considered as significant and *p* < 0.01 as highly significant.

## 3. Results

### 3.1. Physio-Mechanical Analysis of the Coat

Two fractions were distinguished from the tested control samples: cover hair and down hair. The ratio of both fractions was variable and depended on the season and where the animals were kept (Figure 1).

Cover hair in the I-L-S group accounted for about 22% of the sample, and in the III-O-S group, about 13% of the sample. The down hair obtained at that time prevailed over the cover hair and accounted for 78% and 87% of the tested material, respectively. Studies during the II-L-W and IV-O-W periods showed that cover hair accounted for about 13% and 4% of the test material, respectively. As in the case of experiments carried out during the summer, down hair prevailed (87% and 96%, respectively).

Similarly, to the changes taking place in the proportion between cover and down hair, the HTI also changed (Figure 2). The coat of animals kept in laboratory conditions was characterized by much higher HTI values than in rabbits staying in external conditions. The lowest HTI value, and thus the best thermal insulation, was obtained during tests conducted in winter in outdoor conditions (HTI = 0.0435 W/mK).

The results of measurements of the diameter of down and cover hair are collected in Table 4.

The additive of linseed oil ethyl esters did not affect the physicochemical characteristics of termond rabbit cover hair except the cover hair diameter, which in the control group was statistically higher (*p* = 0.0410).

During experiment stages carried out in the laboratory condition, the lower down hair diameter (*p* = 0.0101) was observed compared to down hair diameter obtained from rabbits kept in outdoor condition. Moreover, during summer time, the rabbit hair was characterized by higher down hair breaking tension (*p* = 0.0002), cover hair breaking tension (*p* = 0.0088), and higher cover hair elongation compared to hair obtained from rabbits kept in outdoor condition. However, in both living conditions (laboratory and outdoor) the cover hair diameter was similar. 

In animals kept during the summer time, the lower down hair diameter of rabbit cover hair was observed (*p* = 0.0008), and lower down hair breaking tension (*p* = 0.0053) and lower cover hair elongation (*p* = 0.0017) were compared to hair characteristics obtained from rabbit kept during winter time. However, the time of year did not influence cover hair diameter and cover hair breaking tension. 

Statistical analysis did not show any interaction of experimental factors on the tested parameters.

### 3.2. Histological Analysis of Hair

Histological analysis of the hair showed that the cuticles of down hair were arranged longitudinally to the long axis of the hair. These were cuticles of the flaky, elongated type. The edges of all the cells of the cuticular layer were smooth. The cuticles of down hair were evenly distributed and arranged far from each other. Only one type of down hair was observed in all test animals (Table 5).

It was observed that the administered preparation with a high concentration of omega-3 acids improved the structure of down hair (a clearer drawing, optically greater smoothness of the epithelial-scaly layer).

The cuticular layer of the cover hair of rabbits of the termond white breed was characterized by two types of cell structure, depending on the place of their occurrence along the length of the hair (Table 6). In all cover hairs, the occurrence of the medulla was observed. It was a multicellular core whose cells formed an intermittent ladder pattern (Figure 3). It was shown that the cuticles located closer to the hair root were arranged transversely in relation to the longitudinal axis of the hair. Their shape was characteristic of the type of cuticles referred to as the broadly lobed type. The edges of these cuticles were slightly wavy and delicate irregularities on them could be seen. The distance between the edges of individual cuticles was small and even (Table 7). The cuticular layer located further from the root (closer to the top of the hair) were cells arranged longitudinally to the axis of the hair. The shape of these cells was characteristic of the elongated lobe type. The edges of the cuticles were smooth, devoid of any unevenness. The cuticles were arranged close to each other and partially overlapped (Table 8). 

Analysis of SEM images showed no changes in the histological structure of the cover hair of termond rabbits. No modifications were observed in terms of the structure and arrangement of the cells of the cuticular layer both in the proximal part of the root and in the part closer to the top of the hair. A slightly clearer pattern of cuticles and a lower number of lesions on the surface of the hair after a period of supplementation were observed.

### 3.3. Fatty Acid Profile

Statistical analysis showed a clear effect of the administered preparation on the fatty acid profile in the sebum of coat fur (Table 9).

The additive of linseed oil ethyl esters influenced fatty acids profile in rabbit coat sebum (*p* < 0.01). The concentration of UFA (MUFA and PUFA) were statistically higher in the experimental group and the concentration of SFA was lower in the experimental group (*p* = 0.0014) compared to the control group.

During experiment stages carried out in the outdoor condition the lower concentration of SFA (*p* = 0.0207) and n-6 fatty acids (*p* = 0.0000) were observed compared to laboratory conditions. In contrast to concentration of MUFA, which was lower in samples collected from rabbits kept in laboratory condition (*p* = 0.0111).

Statistical analysis showed the influence of the season on the concentration of n-6 fatty acids, which was higher (*p* = 0.0000) in samples collected during the winter period of the experiment.

Statistical analysis did not show any interaction of experimental factors on the tested parameters. 

In the n-3 acids group, ALA, EPA, and DHA and in the n-6 acids group, LA, GLA, and ALA were determined. In each stage of the study, the content of these acids significantly increased (*p* < 0.01) as a result of linseed oil ethyl esters supplementation (Table 10).

During experiment stages carried out in the outdoor condition, higher concentration of ALA (*p* = 0.0324) and lower concentration of LA (*p* = 0.0000) were obtained compared to laboratory conditions.

Statistical analysis showed the influence of the season on the concentration of n-6 fatty acids (LA and GLA), which was higher (*p* < 0.01) in samples collected during the winter period of the experiment.

In the experimental group (with linseed oil ethyl esters) the increase of ALA content in the sebum of rabbit hair coat was 71% compared to the control group, while EPA and DHA showed an increase of about 425% and 736%, respectively. Supplementation of ethyl esters of linseed oil also had a significant impact on the level of omega-6 acids such as LA (+4%), GLA (+11%), ALA (+28%) in the sebum of rabbit hair coat.

Statistical analysis did not show any interaction of experimental factors on the tested parameters.

## 4. Discussion

The share of various types of hair in the coat, and thus also the density of the coat, is one of the most important parameters determining the quality of the coat [22]. Figure 1 shows clear differences in the percentage of each hair fraction taking into account the season and conditions of maintenance. In animals kept in external conditions (III-O, IV-O), there was much more down hair, which was caused by the instability of weather conditions (ex. different temperatures between day and night), and consequently the need for better thermal insulation of the body. This difference is obvious due to the need for much greater insulation of the coat during low temperatures. In rabbits in laboratory conditions (I-L, II-L), a much larger amount of cover hair was obtained, due to stable, higher temperatures in the room. However, despite unchanged environmental conditions prevailing in the animal house, both in summer and in winter, rabbits underwent a molting process, changing the coat from “summer” to “winter”. Most likely, this was due to the shortening light day, which was the only determinant of the change of season for animals kept indoors.

Hair of animals kept under changing conditions of the external environment (III-O, IV-O) was more susceptible to rupture and damage than that obtained from animals in laboratory conditions (I-L, II-L). In addition, higher breaking stress values observed in the experimental groups compared to the control groups (stages: I-L-S, II-L-W and III-O-S) may suggest a positive effect of supplementation of ethyl esters of linseed oil on the tested cover hair feature, however this was not statistically confirmed.

The heat transfer index is a measure of the heat passing through a sample exposed to thermal radiation. The lower the HTI value, the better the insulator the test material is. In the case of animals, better insulation of the coat means better keeping heat on the animal’s skin, which in turn causes the animal to maintain the desired body temperature. In addition, increased heat protection also means that excess heat from the outside does not pass into the animal’s skin, which in turn protects the body from overheating during hot weather. The coat of animals kept in laboratory conditions (I-L, II-L) was characterized by a much higher heat transfer coefficient than rabbits staying in external conditions (III-O, IV-O), which means weaker insulation of the coat, and consequently weaker thermal protection of the body. This was due to stable environmental conditions in the room, and above all higher and constant temperature and constant air humidity. In external conditions, the HTI value was much lower, i.e., the hair cover showed greater insulating properties and the animals did not freeze, despite different temperature and air humidity values. The lowest HTI value, and thus the best thermal insulation, was obtained during tests conducted in winter in outdoor conditions (HTI = 0.0435 W/mK).

Heat protection is one of the parameters determining the comfort of using fiber products. This feature is influenced by, among others, the type of hair fibers, their structure as well as the properties of the yarn, and the structure of the fabric made from the hair fibers in question [23]. The thermal insulation of materials is important due to the fact that it determines their purpose [24]. The available literature lacks research on the heat-insulating properties of rabbit wool. According to Żyliński [25], the heat protection of wool and woolen materials such as non-woven fabrics and knitted fabrics is in the range of 0.0440–0.0528 W/mK and is lower compared to vegetable fibers. This fact may indicate better insulating properties of materials made of fibers of animal origin. In a study conducted by Bucişcanu [26], thermal conductivity values for sheep’s wool were obtained at the level of 0.037 W/mK. In turn, Hansen et al. [24] and Ye et al. [27] report that the thermal conductivity for wool can be 0.047–0.049 W/mK, depending on the moisture level.

The heat protection of the fabric is due to the insulation of the air between the fibers and the yarn. Fabrics made of straight fiber yarn quickly release heat by conduction when placed next to the skin. On the other hand, hairy fiber fabrics, due to the air insulation between the fabric fibers and the skin, retain body temperature [28]. Studies aimed at assessing and analyzing the thermal comfort of fabrics investigated the relationship between the type of fibers and the composition of fabrics and thermal comfort [28,29,30]. In these studies, it was shown that both the composition of the fibers and the structure of the fabric made of them have a significant impact on the thermal properties and moisture transfer of tested textile materials. It has also been shown that the properties of fibers have an impact on the subjective feelings of users of clothing made from fibers. According to Sirvydas et al. [29], the thermal comfort of the fabric is determined by the thickness, parameters regarding water absorption, and thermal conductivity. In turn, the thermal resistance of clothing as a set of textile materials depends on the thickness and porosity of individual layers, but since the changes in porosity of standard textile materials used in the production of clothing are not large, the total thermal resistance of clothing really depends on the thickness of the material [30].

Hair thickness is one of the most important features that characterize the hair fiber in terms of suitability for further processing [31,32]. The most useful for the production of high-quality yarn are thin, coreless fibers [33,34,35]. The experiment has not proven the effect of administering ethyl esters of linseed oil on the thickness of down hair. Both down and cover hair of animals kept in outdoor conditions were much thicker than down hair of animals in laboratory conditions. Research on the thickness of rabbits’ coat was carried out by among others Khalil et al. [36]. The authors studied the cover of New Zealand and Californian rabbits and in the case of both breeds noted thickness of down hair in the range of 12–18 μm and cover hair in the range of 62–89 μm, depending on the age of the animals—the older the larger the hair diameter. This may explain the statistically higher values of down hair diameter obtained from animals kept in winter (i.e., older rabbits). Taha et al. [37], based on the diameter of the hair follicles, analyzed the diameter of the hair fibers of the Gabali rabbits, New Zealand white and Rex, and obtained results of approx. 35, 48, and 36 μm for primary follicles and 8, 14, and 13 μm for secondary follicles, respectively. In turn, wool derived from Angora rabbits is one of the most delicate animal fibers used in the textile industry. The thickness of down fibers of Angora rabbit wool ranges from approx. 7 to 16 μm, depending on many factors, such as age, gender, and environment [32,38,39], while the thickness of core hair can reach approx. 65 μm [40]. The results obtained in these studies for down hair are at the top of this range, while the results obtained for cover hair are slightly higher. However, it should be remembered that Angora rabbits, unlike the termond rabbits used in the experiment, are a breed typically used in wool production. In addition, according to Taha et al. [37], the most important feature to be taken into account when assessing the cover is its smoothness, which largely depends on the diameter of the fiber. Thick fibers can cause irritation when in contact with the skin. Therefore, an increase in the diameter of the fibre in the case of rabbit cover intended for use in the textile industry is considered an undesirable feature. Beroual et al. [41] conducted a study in which they analyzed the effect of adding flax grain into rabbit feed and rubbing linseed oil into the skin on hair growth and thickness. The authors observed that rubbing linseed oil had an effect on hair thickness, which in the experimental group increased after 4 weeks of the experiment by about 44% compared to the control group. Feeding flax seeds with feed led to different results, in the initial stage it caused a decrease in hair thickness, while an increase in this parameter was observed after 16 weeks of the experiment. However, it is worth noting that giving flax with food as well as rubbing linseed oil into the skin had a beneficial effect on hair growth and mass. The authors suggest that α-linolenic acid (ALA) contained in flaxseed oil and linseed oil may inhibit the activity of 5-α-reductase, the enzyme responsible for converting testosterone to dihydrotestosterone. This hormone causes hair follicle shrinkage and changes in the hair cycle, so inhibition of its formation may explain the beneficial effect of flax on the coat [41].

The breaking force is the force to be applied in order to obtain maximum possible elongation value without the disruption of continuity of fibres of plant or animal origin, including hair. Its study is aimed at determining the tensile strength of the material. The higher the force needed to tear the hair, the more flexible and resistant to damage it is. Breaking stress, like breaking force, is one of the determinants of the quality of an animal hair coat, by determining the elasticity and resistance of hair to mechanical damage. Mengüç et al. [39] report that the hair elongation of the Angora rabbit is in the range of approx. 40–57%, which is definitely higher than the results obtained in their own research. In turn, the elongation values obtained in studies conducted by Wyrostek et al. [42] on the coat of cats was in the range from 10% to 32% depending on the type of hair and the color of the coat (higher values were obtained for dark cover), while in horse hair this parameter ranged from approx. 44% to 55% depending on the breed of horse and type of hair [43]. In turn, the breaking stress values obtained in these studies were about 2–3 times lower compared to the coat of cats or fur animals, in which this parameter was at the level of approx. 4–7 kg/mm^2^, and definitely lower than the values obtained for the cover of dogs or sheep (about 15 kg/mm^2^) [44,45].

The physio-mechanical properties of hair fibers are extremely important and determine suitability in the textile industry and purpose, testify to the condition of the hair, and thus indirectly also to the condition and health of animals [43,46,47]. One of the most important factors affecting the strength of fibers is air humidity: at higher values the fiber is more stretchable, because water acts as a plasticizing agent, while dry fibers, thanks to hydrogen bonds, are resistant to elongation. A similar effect was observed for temperature, as it increases, the hair fiber is weaker and more prone to stretching. An additional factor affecting the reduction of fiber strength are various types of acidic and alkaline substances [48]. The physical characteristics of rabbit hair were decisively influenced by the environmental conditions in which the animals lived and the time of year.

The hair of all animal species is characterized by a similar cellular structure, consisting of the medulla (which is not always found in down hair), a cortical layer, and a cuticular layer. However, the detailed structure of the individual layers is a genetically determined feature, characteristic of each species. The most diverse is the cuticular layer. Thanks to such characteristics as the shape and arrangement of cuticles, the appearance of the edges of cuticles and their distance from each other, it is possible to identify the species of animal, even after many years, which is commonly used, for example, in forensic science and archaeology. The arrangement and shape of the cover cells is also a characteristic feature of a given animal species [49,50]. The histological structure in terms of the arrangement of cuticles of the outer layer of rabbit hair, as well as the medulla obtained in own research, is consistent with the breed standard and data presented in the literature [51,52].

Ethyl esters of linseed oil are a rich source of many valuable fatty acids, especially those from the omega-3 group [19]. In order to investigate the effect of the administered preparation on the quality of rabbits’ coat, the profile of fatty acids contained in sebum covering the hair was analyzed. The fat found on the hair is produced by the sebaceous glands and covers each hair with a thin, waterproof layer. Its main role is to protect hair from damage and water loss, as well as their nutrition [53]. Sinclair et al. [54] suggest that α-linolenic acid can enter the surface of hair fibers through the sebaceous glands and has a protective function against damage to the hair by water, light, or other harmful factors. Studies have also shown that ALA can be a factor in improving the growth of the coat. It was also observed that a diet low in ALA and rich in LA acid caused skin changes and hair loss. Changes in the ratio of SFA to UFA, especially an increase in the share of omega-3 acids and a decrease in the ratio of n-6: n-3 acids are therefore important in the context of improving protective properties by better moisturizing the hair surface [55,56]. MUFA, i.e., linoleic acid and α-linolenic acids, belong to the so-called essential fatty acids, which the body is not able to synthesize de novo and therefore must be supplied with the diet. These acids are precursors of subsequent long-chain fatty acids, which are biosynthesized with the participation of enzymes in processes involving elongation and desaturation. Both precursors compete for the same enzyme, D6-desaturase, which has a greater affinity for linolenic acid, which is why an adequate supply of α-linolenic acid is so important [57,58]. However, a controversial issue is the effectiveness of epa synthesis, and especially DHA from α-linolenic acid, which according to literature data is low and amounts to only about 6% and 3.4%, respectively [57,59,60,61,62]. This phenomenon is associated with the final stage of the DHA biosynthesis pathway, β-oxidation of C24:6n-3, which involves translocation between the endoplasmic reticulum and the peroxisomes of DHA and its precursor (C24:6n-3) [57,63].

Research in the field of nutritional modification of the fatty acid profile in rabbits and other livestock species was carried out mainly in terms of the quality of their meat [64,65,66], which is in line with current trends in functional food. However, it is also worth bearing in mind the aspect of animal health related to the activity of fatty acids from the omega-3 family, especially ALA as a precursor of EPA and DHA. In this study, rabbits were used as model animals, in the aspect of studies on the effect of omega-3 fatty acids on the state of the hair cover of fur animals [67]. As a result of supplementation, significant, beneficial changes in the fatty acid profile in hair sebum were observed. A significant increase in omega-3 acids, and a significant decrease in the ratio of omega-6 to omega-3 acids was observed.

## 5. Conclusions

Rabbits bred for fur should be kept in outdoor conditions, as is the case with other fur animals such as common and arctic foxes or American mink. Histological analysis of the hair of termond rabbits showed a variation in the structure of the cuticle depending on the type of hair. In the case of cover hair, the structure of the cuticular layer was also differentiated depending on the place of hair examination. The applied preparation of ethyl esters of linseed oil had a positive effect on the histological image of hair visible in a clearer drawing of cuticles and significantly higher strength. As a result of supplementation, significant, beneficial changes in the fatty acid profile in hair sebum were observed. There was a significant increase in omega-3 acids, and a significant decrease in the ratio of omega-6 to omega-3 acids.

## Figures and Tables

**Figure 1 animals-12-00971-f001:**
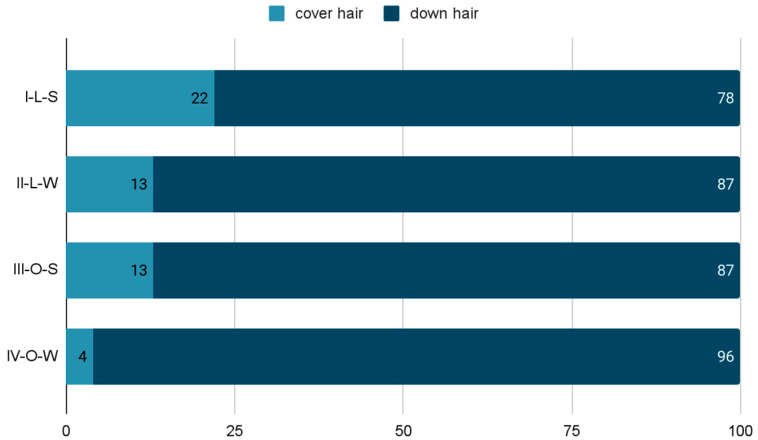
The proportion of cover and down hair (%) in the hair coat test—control group (L—laboratory conditions; O—external conditions; S—summer; W—winter).

**Figure 2 animals-12-00971-f002:**
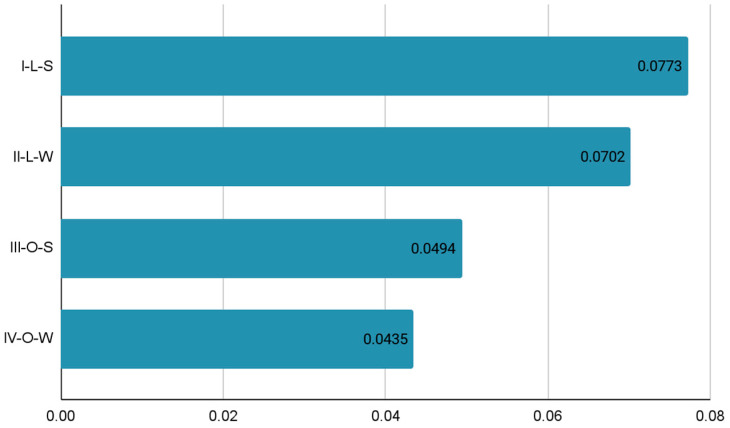
Heat transfer coefficient (W/mK) of rabbit coat—control group (L—laboratory conditions; O—external conditions; S—summer; W—winter).

**Figure 3 animals-12-00971-f003:**
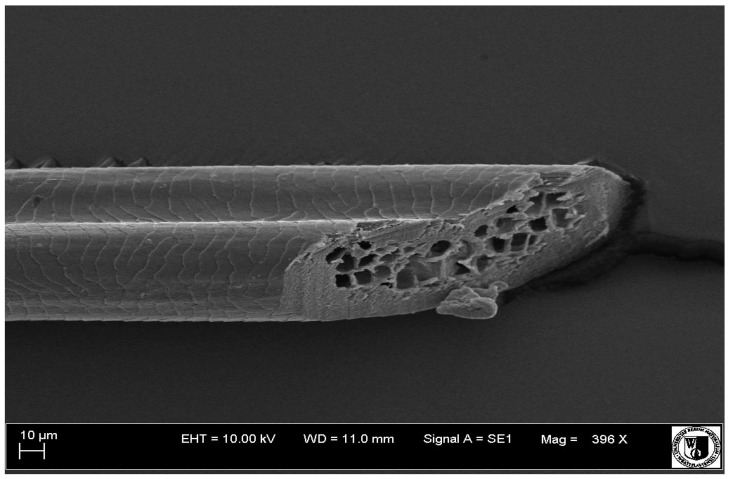
Cross-section of the cover hair of the termond rabbits—visible cells of the hair core.

**Table 1 animals-12-00971-t001:** Ingredients (g/kg) and chemical composition of granules (g/kg of dry matter).

Ingredients		Composition	
Alfalfa	205	Crude protein	166.3
Grass mixture	135	Crude fiber	148.2
Wheat bran	230	Crude fat	21.4
Dried molasses beet pulp	120	Crude ash	84.2
Beet molasses	100	Calcium	11.2
Sunflower post-extraction meal	60	Sodium	2.6
Rapeseed post-extraction meal	50	Phosphorus	9.7
Corn	40		
Post-extraction soybean meal (toasted)	20		
Mineral-vitamin supplement *	40		

* calcium carbonate (20 g/kg), monocalcium phosphate (2 g/kg), sodium chloride (5 g/kg), sodium bicarbonate (25 g/kg), vit. A (8000 IU/kg), vit. D3 (1200 IU/kg), vit. E (25 IU/kg), vit. K (0.4 mg), vit. B1 (0.4 mg), vit. B2 (3.2 mg), vit. B6 (0.4 mg), vit. B12 (12 mg), biotin (80 mg), folic acid (0.45 mg), nicotinic acid (16 mg), pantothenic acid (6 mg).

**Table 2 animals-12-00971-t002:** Average fatty acid content of basic feed, hay and ethyl esters of PUFAs obtained from linseed oil.

Acid	Hay	Feed	Linseed Oil Ethyl Esters	Acid	Hay	Feed	Linseed Oil Ethyl Esters
Saturated fatty acids				Unsaturated fatty acids			
C6:0	0.57	-	-	C14:1	-	0.07	-
C8:0	0.66	0.02	-	C16:1	2.33	0.32	-
C10:0	0.64	-	-	C17:1	-	0.05	-
C12:0	1.26	0.03	-	C18:1	-	-	16.73
C14:0	2.4	0.12	-	C18:2n-6c	16.82	50.17	16.68
C15:0	-	0.04	-	C18:2n-6t	17.66	21.32	-
C16:0	28.83	15.39	4.44	C18:3n-6	2.41	-	-
C17:0	-	0.1	-	C18:3n-3	5.3	5.9	58.71
C18:0	4.98	4.24	3.43	C20:4n-6	-	0.07	-
C20:0	-	0.38	-	C20:5n-3	1.75	-	-
				C22:6n-3	-	0.56	-

**Table 3 animals-12-00971-t003:** Average content of omega-6 and omega-3 acids, and n-6/n-3 ratio in feed and esters.

	n-6 Acids	n-3 Acids	n-6/n-3
Basic feed	72.03%	6.46%	11.15
Linseed oil ethyl esters	16.68%	58.71%	0.28
Acid content in 5 mL of esters	0.71 g	2.50 g	0.28
Content in feed with added ethyl esters	3.02 g	2.71 g	1.11

**Table 4 animals-12-00971-t004:** Impact of environmental factors (living conditions, seasons) and applied supplementation on chosen physicochemical characteristics of termond rabbit cover hair.

	Down Hair Diameter (μm)	Cover Hair Diameter (μm)	Down Hair Breaking Tension (N)	Cover Hair Breaking Tension (kg/mm^2^)	Cover Hair Elongation (%)
	mean ± sd	mean ± sd	mean ± sd	mean ± sd	mean ± sd
I-L-S C	13.80 ± 0.51	73.54 ± 8.19	0.26 ± 0.02	1.69 ± 0.50	30.01 ± 1.90
I-L-S E	13.75 ± 0.58	62.48 ± 6.49	0.23 ± 0.03	2.12 ± 0.62	32.41 ± 3.27
II-L-W C	15.07 ± 0.66	68.52 ± 1.94	0.26 ± 0.02	1.92 ± 0.08	32.12 ± 0.35
II-L-W E	14.54 ± 0.52	63.11 ± 2.39	0.24 ± 0.01	2.13 ± 0.02	34.77 ± 1.72
III-O-S C	14.36 ± 0.51	66.25 ± 4.91	0.19 ± 0.01	1.39 ± 0.04	27.59 ± 3.37
III-O-S E	14.76 ± 0.10	67.56 ± 1.92	0.20 ± 0.01	1.55 ± 0.08	27.88 ± 1.10
IV-O-W C	15.28 ± 0.57	69.42 ± 0.68	0.23 ± 0.01	1.82 ± 0.10	31.02 ± 0.72
IV-O-W E	15.14 ± 0.31	68.93 ± 1.33	0.24 ± 0.00	1.64 ± 0.24	32.15 ± 0.87
Additive					
C	14.63 ± 0.78	69.43 ^a^ ± 4.99	0.23 ± 0.03	1.71 ± 0.30	30.19 ± 2.43
E	14.55 ± 0.64	65.52 ^b^ ± 4.26	0.23 ± 0.02	1.86 ± 0.40	31.80 ± 3.09
Condition					
L	14.29 ^a^ ± 0.75	66.91 ± 6.60	0.25 ^A^ ± 0.03	1.96 ^A^ ± 0.39	32.33 ^A^ ± 2.50
O	14.89 ^b^ ± 0.52	68.04 ± 2.67	0.22 ^B^ ± 0.02	1.60 ^B^ ± 0.20	29.66 ^B^ ± 2.60
Season					
S	14.17 ^A^ ± 0.59	67.46 ± 6.49	0.22 ^A^ ± 0.03	1.69 ± 0.44	29.47 ^A^ ± 2.99
W	15.01 ^B^ ± 0.54	67.50 ± 3.04	0.24 ^B^ ± 0.02	1.88 ± 0.22	32.52 ^B^ ± 1.69
*p*-value					
Additive	0.6883	0.0410	0.1127	0.2310	0.0618
Condition	0.0101	0.5304	0.0002	0.0088	0.0045
Season	0.0008	0.9818	0.0053	0.1351	0.0017
Interaction	0.9423	0.3058	0.7036	0.8086	0.8579

Experimental factor: Additive—addition of linseed oil ethyl esters (C—control or E—experimental), Condition—animal living conditions (L—laboratory or O—outdoor cage), Season—season of experiment (S—summer or W—winter), Interaction—interaction between all factors; ^A, B^—highly significant differences at the level of *p* < 0.01; ^a, b^—significant differences at the level of *p* < 0.05.

**Table 5 animals-12-00971-t005:** SEM image of down hair of termond rabbits.

SEM Image of the Cuticular Layer of Down Hair of Rabbits of Termond White Breed	
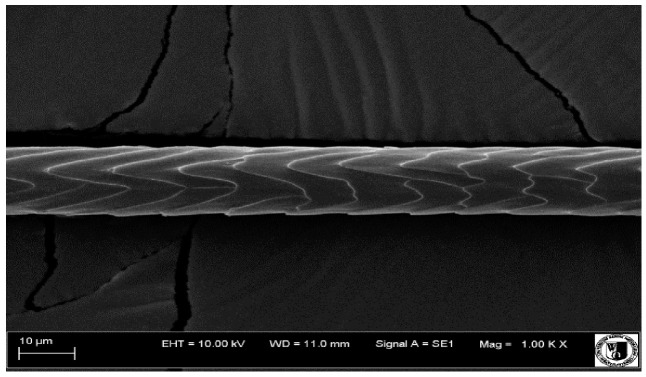	
Period I-L-S	
Before supplementation	After supplementation
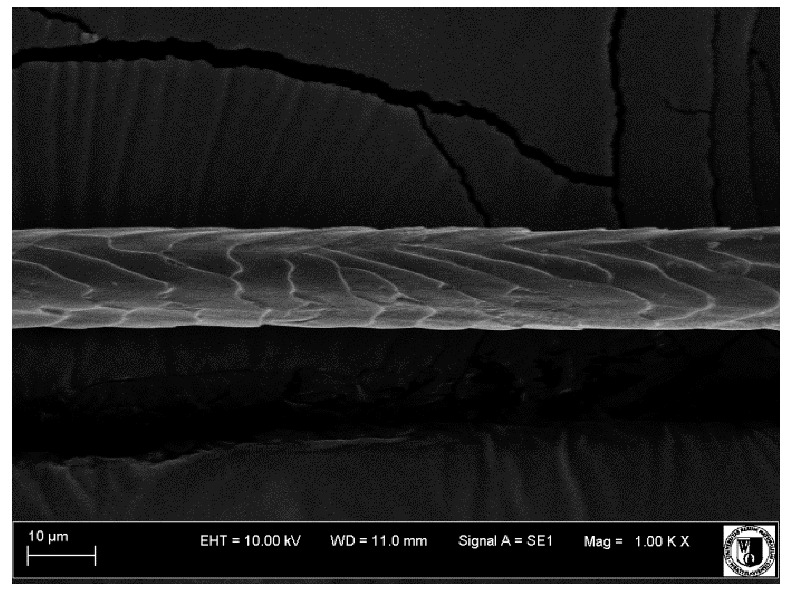	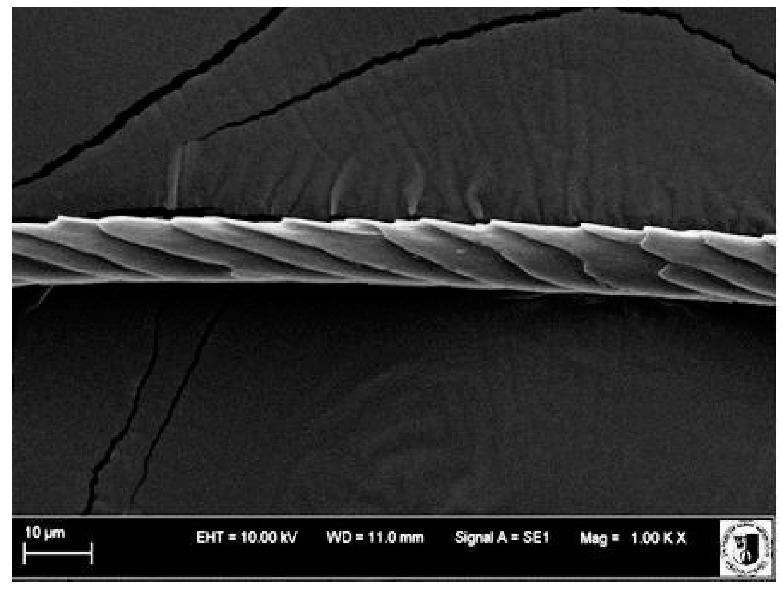
Period II-L-W	
Before supplementation	After supplementation
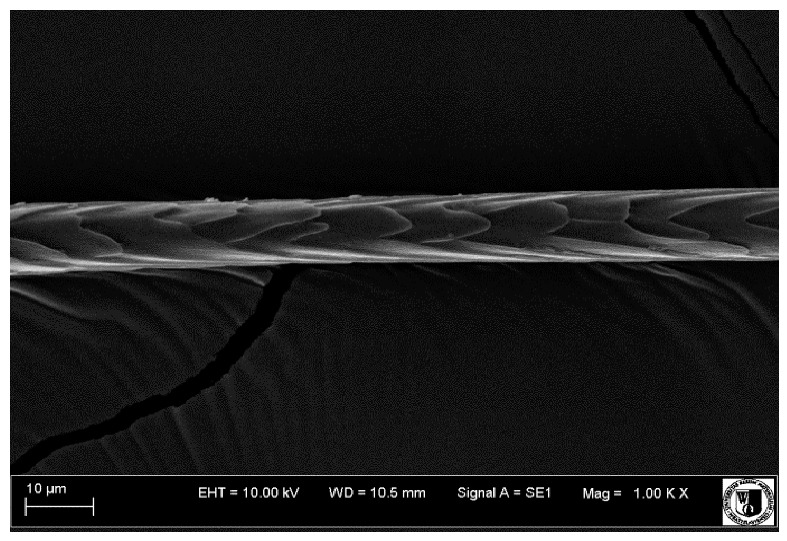	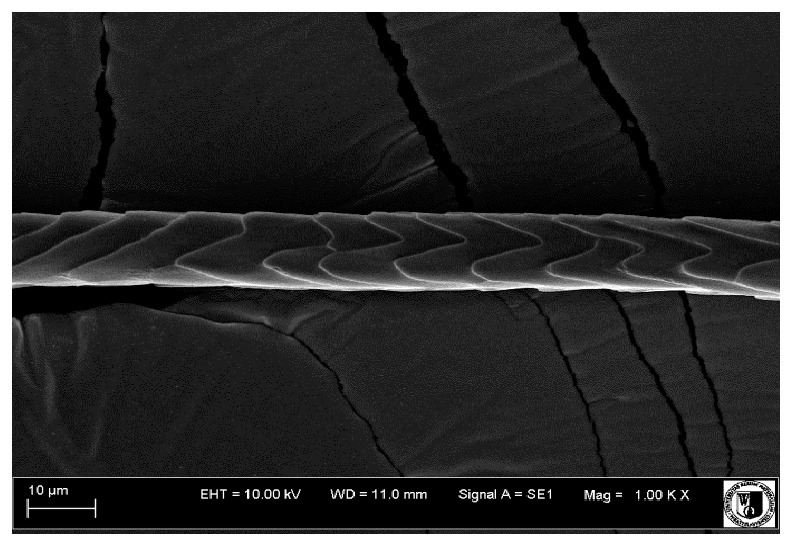
Period III-O-S	
Before supplementation	After supplementation
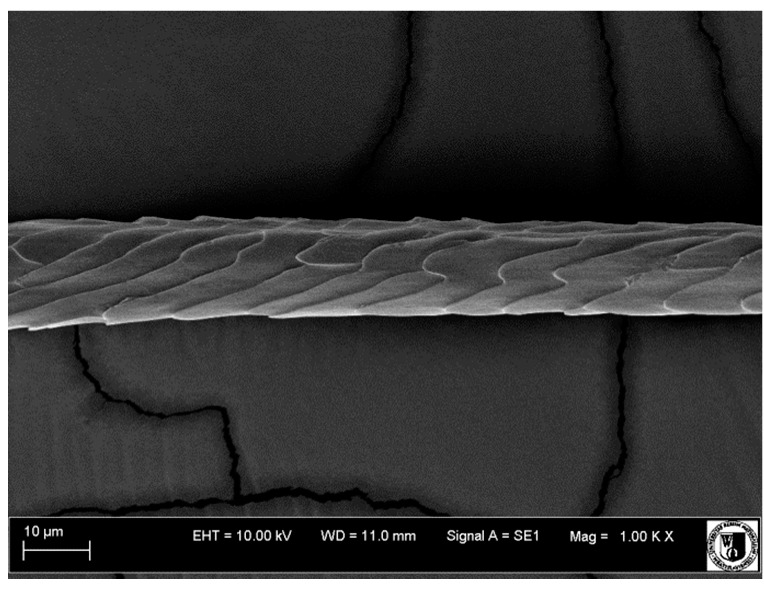	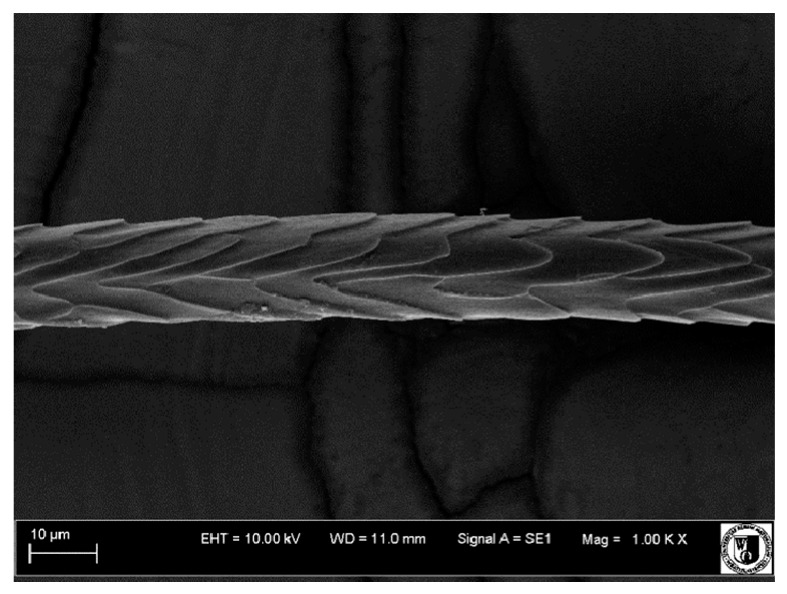
Period IV-O-W	
Before supplementation	After supplementation
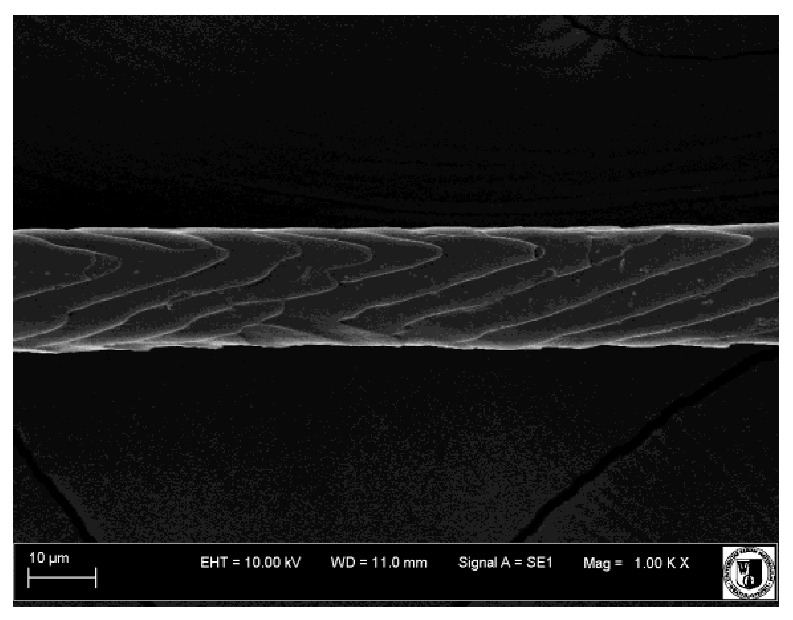	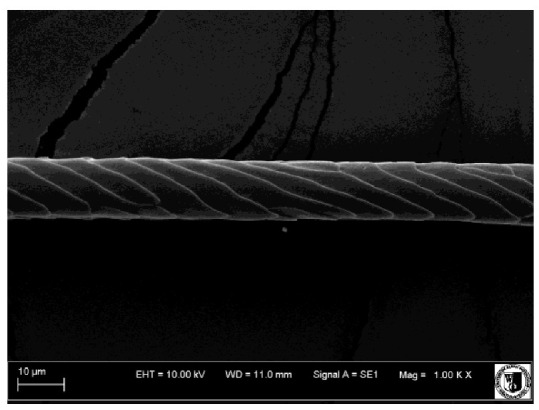

**Table 6 animals-12-00971-t006:** SEM image of the cuticular lDayer of the cover hair of termond rabbits.

The Part Located Closer to the Root.	The Part Located Further from the Root.
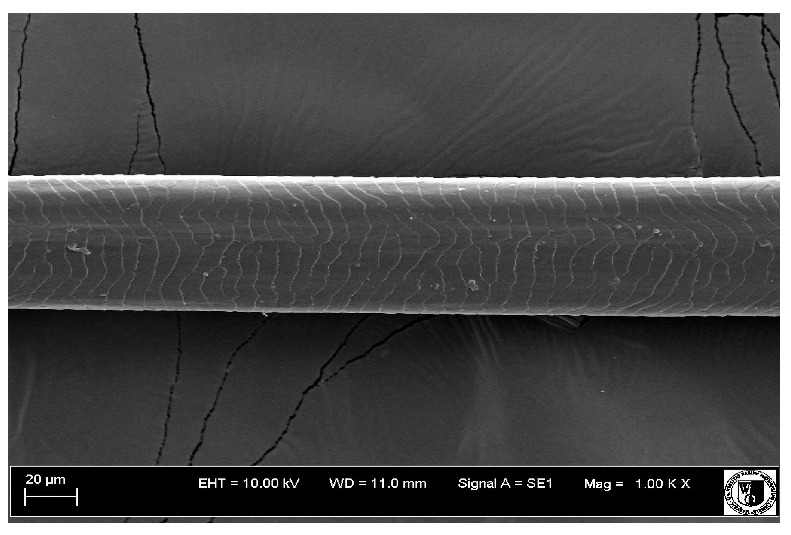	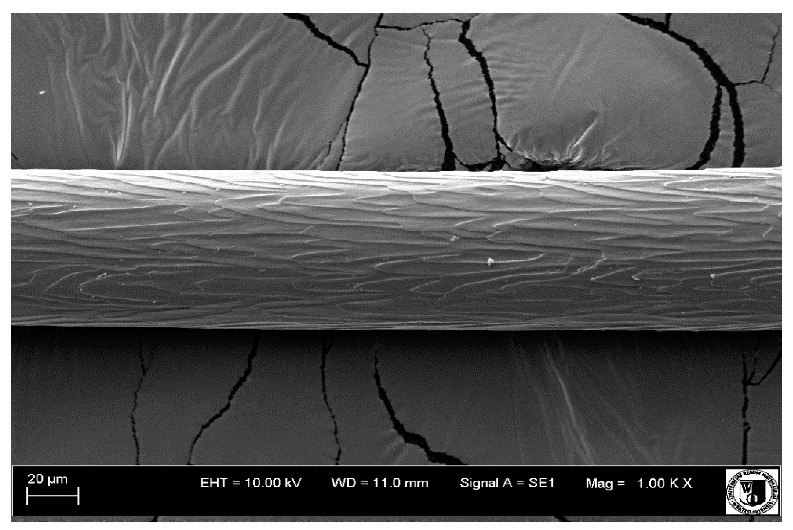

**Table 7 animals-12-00971-t007:** SEM image of cover hair (closer part) termond rabbits.

Period I-L-S	
Before supplementation	After supplementation
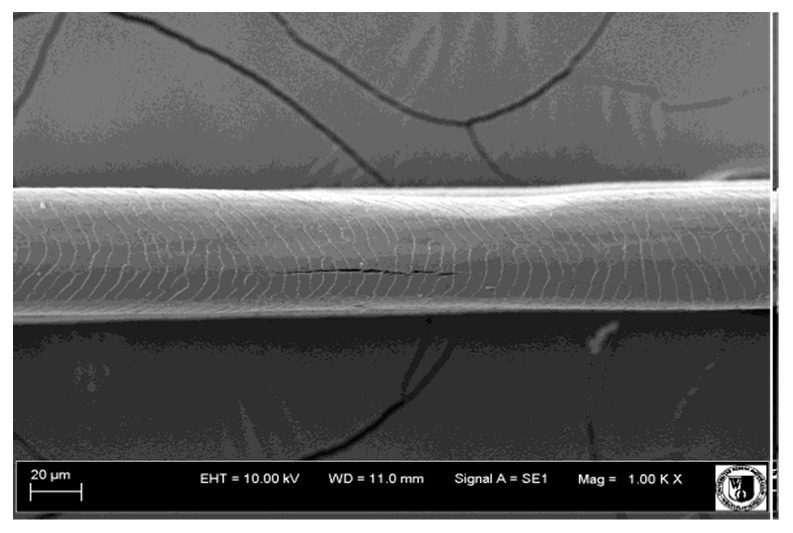	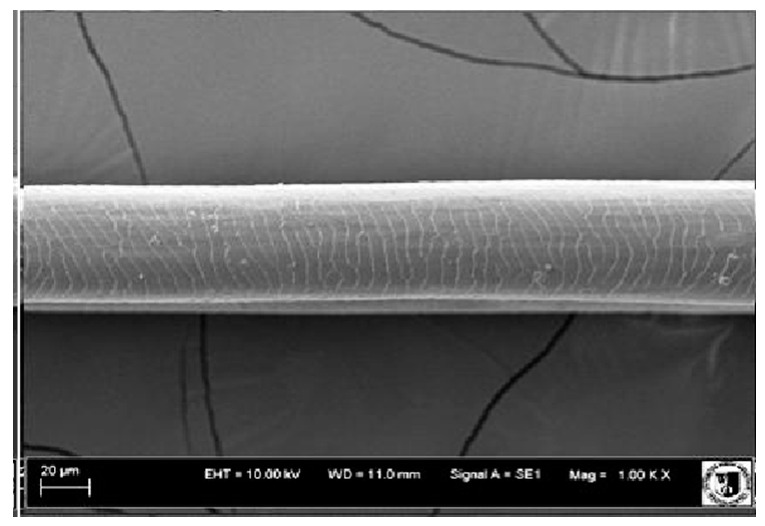
Period II-L-W	
Before supplementation	After supplementation
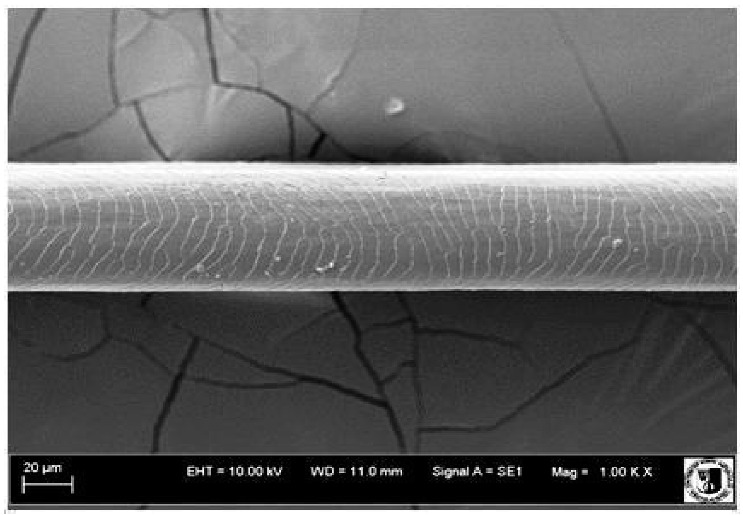	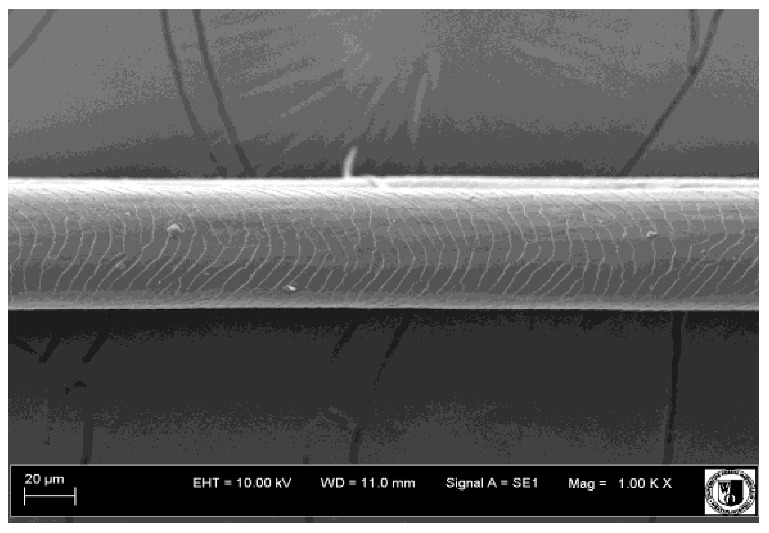
Period III-O-S	
Before supplementation	After supplementation
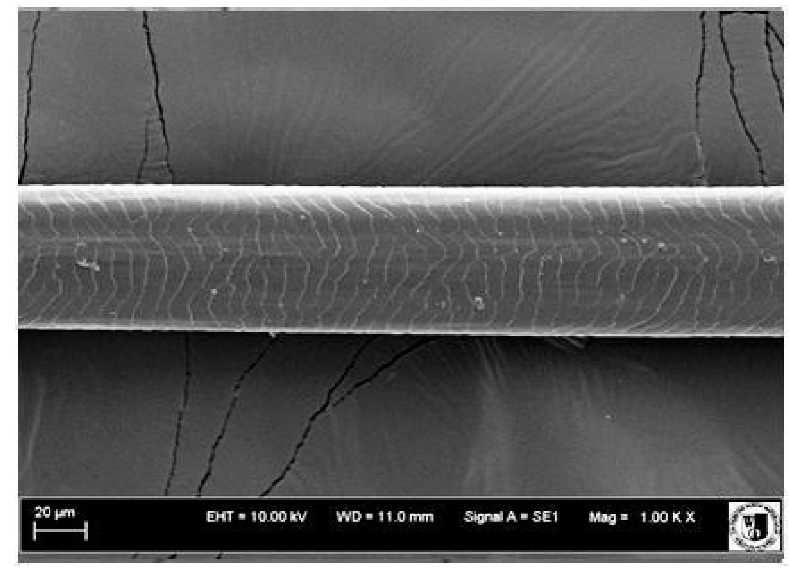	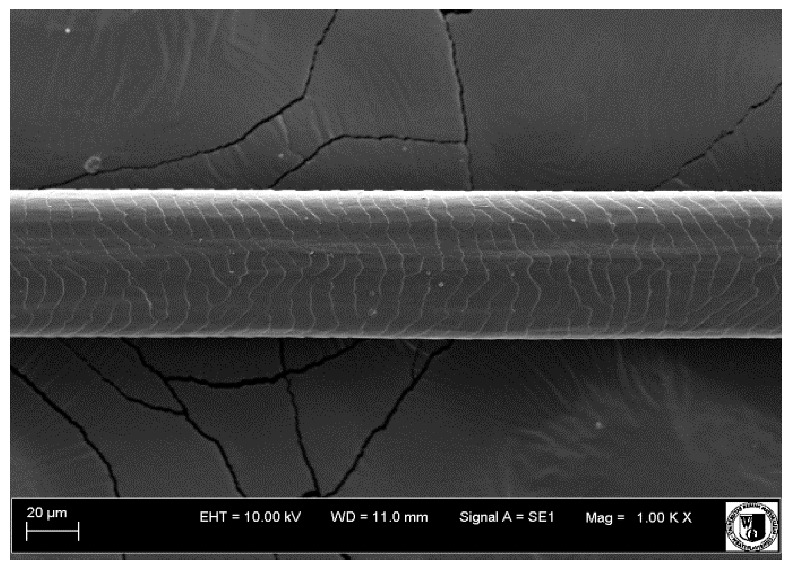
Period IV-O-W	
Before supplementation	After supplementation
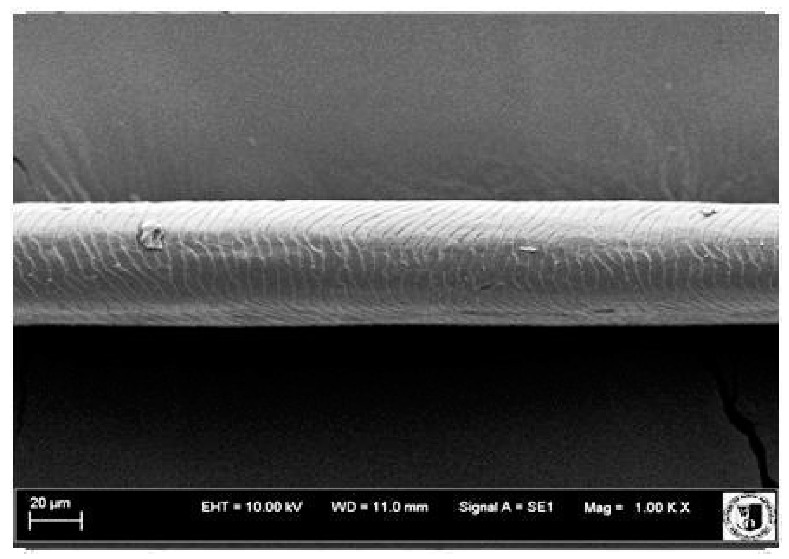	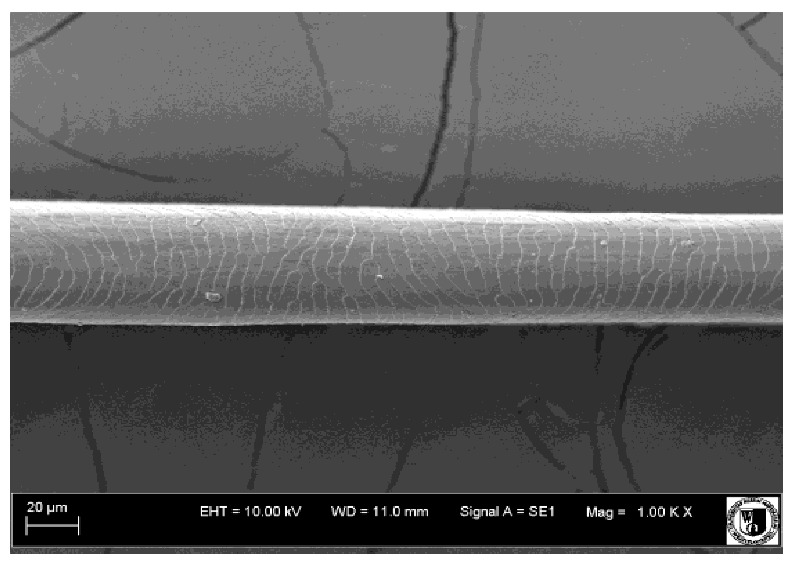

**Table 8 animals-12-00971-t008:** SEM image of cover hair (further part) of termond rabbits before.

Period I-L-S	
Before supplementation	After supplementation
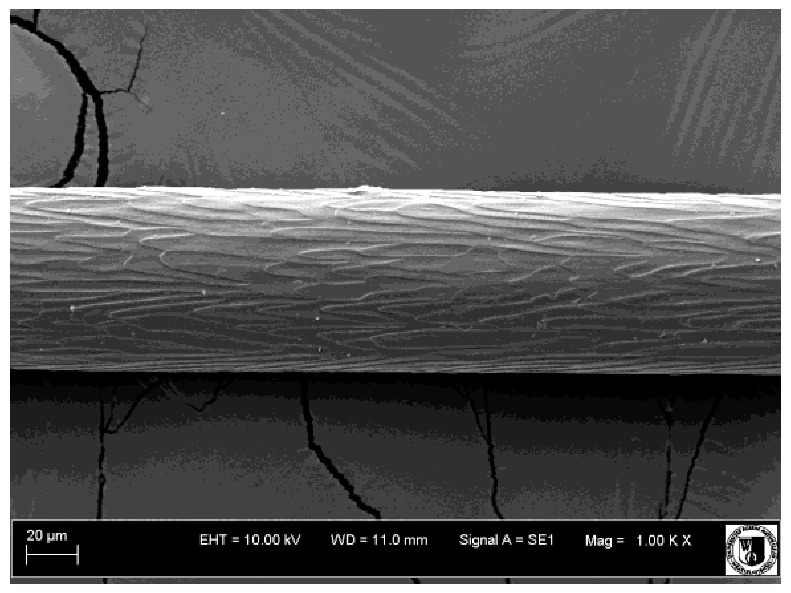	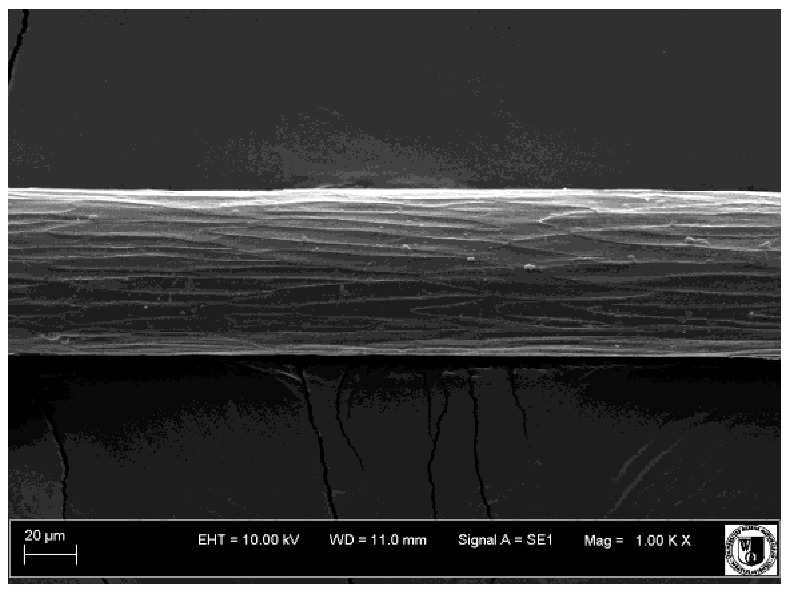
Period II-L-W	
Before supplementation	After supplementation
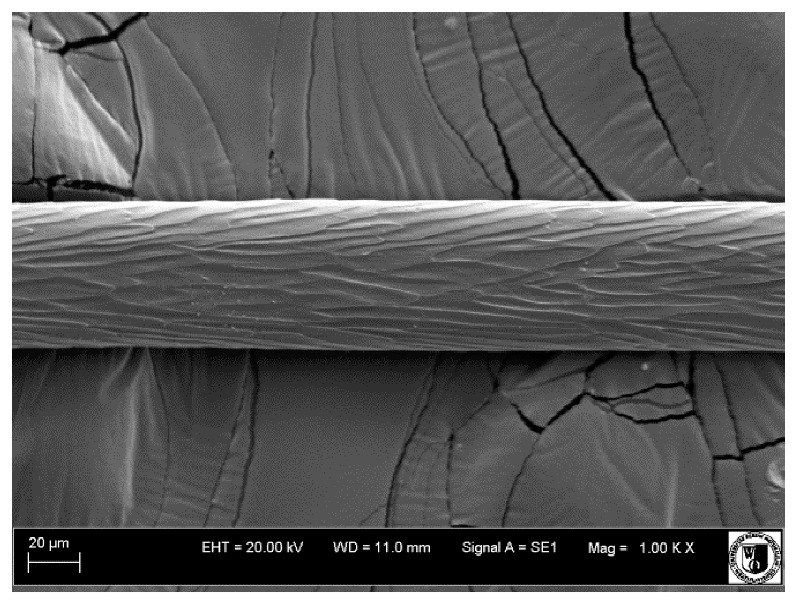	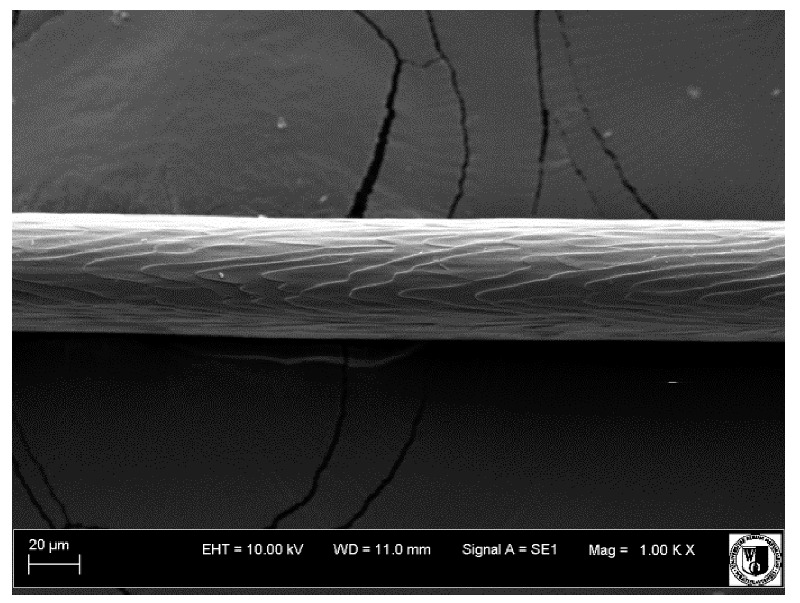
Period III-O-S	
Before supplementation	After supplementation
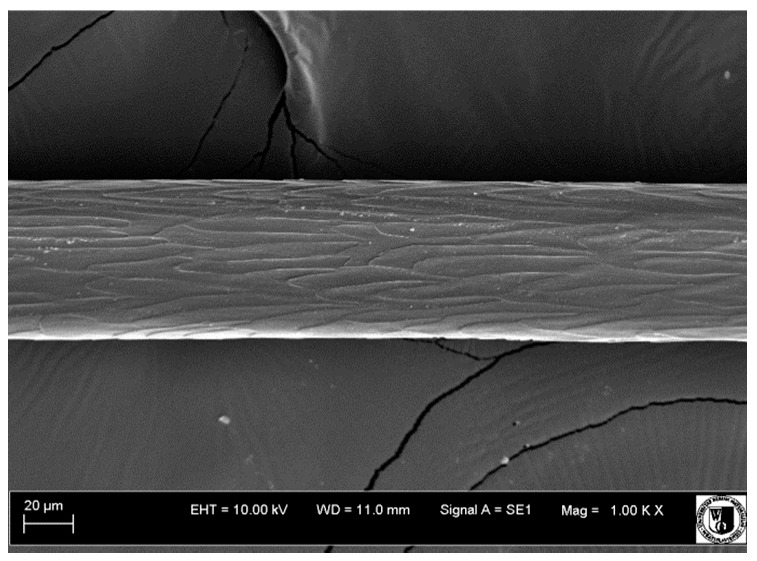	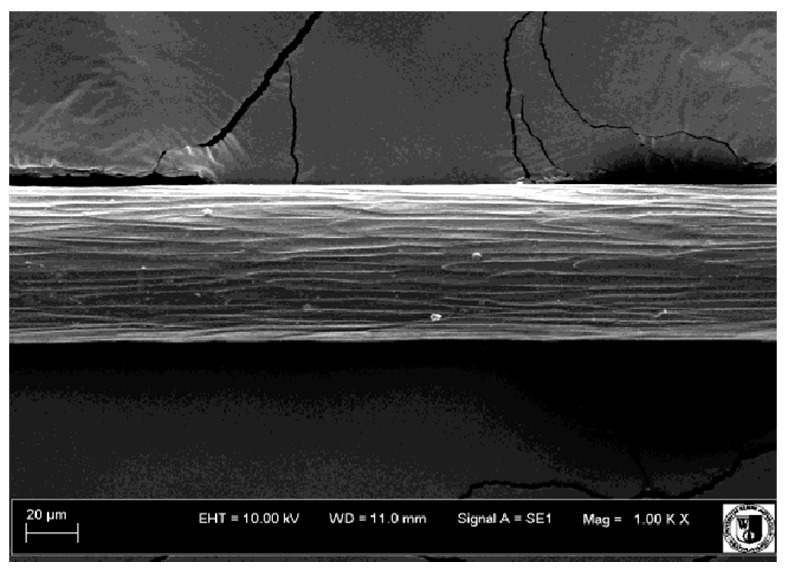
Period IV-O-W	
Before supplementation	After supplementation
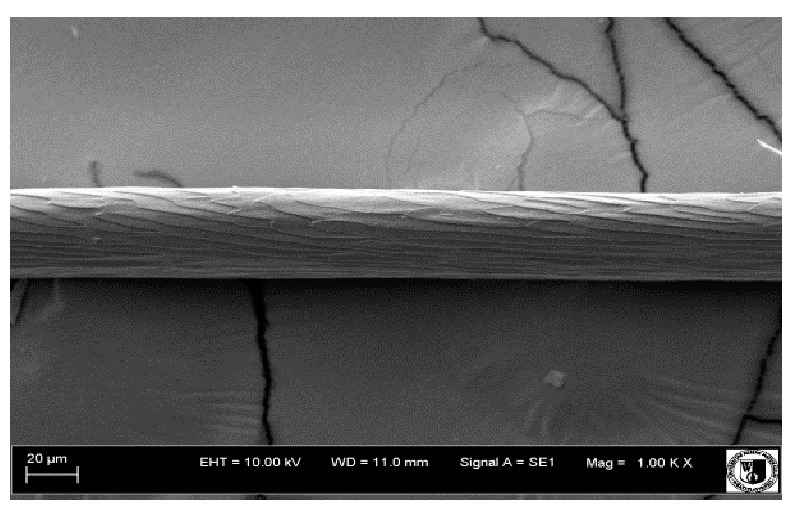	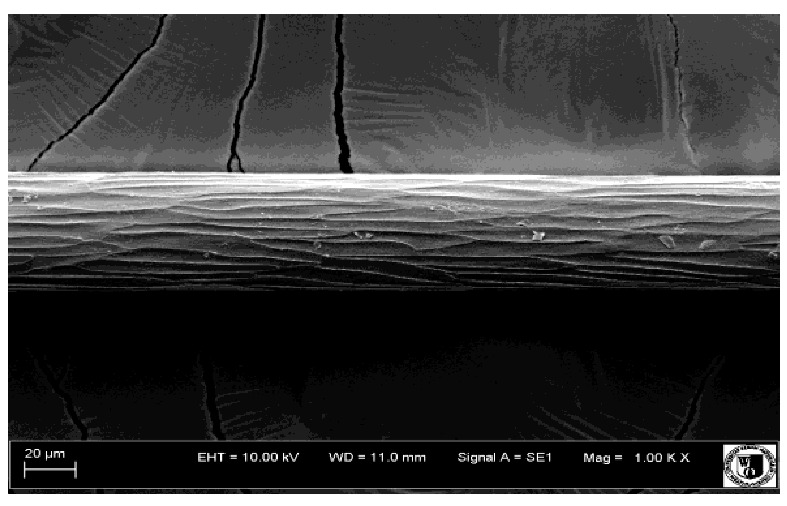

**Table 9 animals-12-00971-t009:** Fatty acid profile in rabbit coat sebum.

	SFA	UFA	MUFA	PUFA	N-3	N-6	N-6/N-3
	mean ± sd	mean ± sd	mean ± sd	mean ± sd	mean ± sd	mean ± sd	mean ± sd
I-L-S C	46.95 ± 0.34	47.90 ± 0.40	28.30 ± 0.22	19.60 ± 0.26	1.02 ± 0.03	17.52 ± 0.21	17.21 ± 0.31
I-L-S E	44.79 ± 1.15	51.94 ± 3.69	29.51 ± 1.32	22.42 ± 2.38	2.94 ± 1.72	18.47 ± 0.64	9.47 ± 8.09
II-L-W C	48.58 ± 0.56	48.61 ± 0.16	27.41 ± 0.11	21.20 ± 0.13	1.53 ± 0.05	19.01 ± 0.11	12.49 ± 0.43
II-L-W E	45.51 ± 2.68	52.97 ± 3.75	28.98 ± 1.24	23.99 ± 2.52	3.38 ± 1.68	19.93 ± 0.88	7.43 ± 4.70
III-O-S C	46.00 ± 0.27	48.93 ± 0.51	28.92 ± 0.29	20.01 ± 0.24	1.69 ± 0.05	16.60 ± 0.12	9.91 ± 0.23
III-O-S E	41.55 ± 4.26	54.08 ± 4.68	30.68 ± 1.67	23.40 ± 3.02	4.06 ± 2.15	17.40 ± 0.60	5.73 ± 4.10
IV-O-W C	46.72 ± 0.24	48.90 ± 0.10	29.00 ± 0.07	19.91 ± 0.13	1.49 ± 0.01	17.40 ± 0.06	11.73 ± 0.10
IV-O-W E	41.86 ± 3.96	53.88 ± 4.73	30.39 ± 1.46	23.49 ± 3.27	4.11 ± 2.28	18.06 ± 0.72	6.20 ± 4.80
Additive							
C	47.06 ^A^ ± 1.04	48.59 ^A^ ± 0.52	28.41 ^A^ ± 0.68	20.18 ^A^ ± 0.66	1.43 ^A^ ± 0.26	17.63 ^A^ ± 0.92	12.84 ^A^ ± 2.83
E	43.43 ^B^ ± 3.32	53.22 ^B^ ± 3.72	29.89 ^B^ ± 1.41	23.32 ^B^ ± 2.48	3.62 ^B^ ± 1.76	18.46 ^B^ ± 1.15	7.21 ^B^ ± 5.04
Condition							
L	46.46 ^a^ ± 1.98	50.35 ± 3.18	28.55 ^a^ ± 1.13	21.80 ± 2.24	2.22 ± 1.44	18.73 ^A^ ± 1.03	11.65 ± 5.54
O	44.03 ^b^ ± 3.49	51.45 ± 3.88	29.75 ^b^ ± 1.26	21.70 ± 2.63	2.84 ± 1.87	17.36 ^B^ ± 0.67	8.39 ± 3.76
Season							
S	44.82 ± 2.85	50.71 ± 3.61	29.35 ± 1.13	21.36 ± 2.34	2.43 ± 1.69	17.50 ^A^ ± 0.79	10.58 ± 5.82
W	45.67 ± 3.29	51.09 ± 3.56	28.95 ± 1.37	22.15 ± 2.48	2.63 ± 1.70	18.60 ^B^ ± 1.11	9.46 ± 4.02
*p*-value							
Additive	0.0014	0.0017	0.0027	0.0014	0.0014	0.0012	0.0033
Condition	0.0207	0.3869	0.0111	0.9024	0.2944	0.0000	0.0629
Season	0.3842	0.7625	0.3427	0.3486	0.7347	0.0000	0.5030
Interaction	0.8958	0.9218	0.6673	0.9455	0.8945	0.9133	0.5447

Experimental factor: Additive—addition of linseed oil ethyl esters (control or experimental), Condition—animal living conditions (laboratory or outdoor cage), Season—season of experiment (summer or winter), Interaction—interaction between factors; ^A, B^—highly significant differences at the level of *p* < 0.01; ^a, b^—significant differences at the level of *p* < 0.05.

**Table 10 animals-12-00971-t010:** Content of omega 3 and omega 6 fatty acids in the sebum of rabbit hair coat.

	Omega 3 Acids			Omega 6 Acids		
	ALA	EPA	DHA	LA	GLA	ALA
	mean ± sd	mean ± sd	mean ± sd	mean ± sd	mean ± sd	mean ± sd
I-L-S C	0.79 ± 0.02	0.13 ± 0.00	0.10 ± 0.01	16.90 ± 0.22	0.32 ± 0.01	0.29 ± 0.02
I-L-S E	1.31 ± 0.50	0.61 ± 0.44	1.01 ± 0.79	17.76 ± 0.79	0.44 ± 0.08	0.34 ± 0.02
II-L-W C	1.33 ± 0.05	0.10 ± 0.01	0.10 ± 0.01	18.10 ± 0.11	0.59 ± 0.02	0.32 ± 0.03
II-L-W E	1.99 ± 0.62	0.56 ± 0.41	0.84 ± 0.53	18.88 ± 0.74	0.65 ± 0.02	0.41 ± 0.13
III-O-S C	1.44 ± 0.05	0.13 ± 0.00	0.12 ± 0.01	15.92 ± 0.11	0.43 ± 0.02	0.24 ± 0.02
III-O-S E	2.41 ± 0.93	0.70 ± 0.51	0.95 ± 0.62	16.58 ± 0.52	0.45 ± 0.02	0.37 ± 0.06
IV-O-W C	1.26 ± 0.01	0.12 ± 0.01	0.11 ± 0.01	16.57 ± 0.06	0.52 ± 0.01	0.31 ± 0.01
IV-O-W E	2.57 ± 1.13	0.65 ± 0.47	0.89 ± 0.52	17.15 ± 0.65	0.55 ± 0.03	0.37 ± 0.06
Additive						
C	1.21 ^A^ ± 0.26	0.12 ^A^ ± 0.02	0.11 ^A^ ± 0.01	16.87 ^A^ ± 0.83	0.47 ^A^ ± 0.11	0.29 ^A^ ± 0.04
E	2.07 ^B^ ± 0.87	0.63 ^B^ ± 0.39	0.92 ^B^ ± 0.60	17.59 ^B^ ± 1.06	0.52 ^B^ ± 0.09	0.37 ^B^ ± 0.07
Condition						
L	1.36 ^a^ ± 0.56	0.35 ± 0.05	0.51 ± 0.06	17.91 ^A^ ± 0.88	0.50 ± 0.14	0.34 ± 0.07
O	1.92 ^b^ ± 0.87	0.40 ± 0.04	0.52 ± 0.06	16.55 ^B^ ± 0.58	0.49 ± 0.05	0.32 ± 0.07
Season						
S	1.49 ± 0.76	0.39 ± 0.04	0.55 ± 0.06	16.79 ^A^ ± 0.81	0.41 ^A^ ± 0.07	0.31 ± 0.06
W	1.79 ± 0.78	0.36 ± 0.03	0.49 ± 0.05	17.67 ^B^ ± 1.02	0.58 ^B^ ± 0.05	0.35 ± 0.05
*p*-value						
Additive	0.0024	0.0013	0.0012	0.0025	0.0011	0.0035
Condition	0.0324	0.7050	0.9715	0.0000	0.3874	0.4944
Season	0.2332	0.7798	0.7783	0.0005	0.0000	0.0788
Interaction	0.8457	0.9653	0.8770	0.9935	0.1919	0.2421

Experimental factor: Additive—addition of linseed oil ethyl esters (control or experimental), Condition—animal living conditions (laboratory or outdoor cage), Season—season of experiment (summer or winter), Interaction—interaction between factors; ^A, B^—highly significant differences at the level of *p* < 0.01; ^a, b^—significant differences at the level of *p* < 0.05.

## Data Availability

Not applicable.

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
