# Peer review of "The Effect of the Season, the Maintenance System and the Addition of Polyunsaturated Fatty Acids on Selected Biological and Physicochemical Features of Rabbit Fur"

_animals, 2022, doi:10.3390/ani12080971_

Round 1

Reviewer 1 Report

In general, it is a very interesting study. It is better for the authors to use more concise language and more intuitive graphs to show the amazing results, rather than too many numbers in the tables and a lot of descriptions of the tables.

- Quality of English: In the manuscript, some language editing is required to correct the grammar and syntax errors. Some expressions need to be more accurate and concise to avoid confusing readers.

- Abstract: More information on methods and results should be provided here.

- Statistical analysis: It was a three-factor study, and each factor with two levels. Therefore, the statistical analysis methods need to be explained in detail. I don’t think the sample size was big enough to use standard deviation, because the size of every single sample was just 8.

- Feeding: The authors need to provide the formulation of the basal diet, as this is related to the performance of the rabbit fur.

- Fig. 1: Which group did the data come from in every stage?control (K) or experimental (D)? And the same question for Fig. 2.

- Table 3: What does the spiral symbol mean?

- 3.2. Histological analysis of hair: I think you can organize the pictures better for easy reading and understanding.

Author Response

Replies to reviewer - animals-1658374

The authors would like to thank the reviewers for all comments and suggestions which may improve manuscript quality. The authors made changes to the text in line with the reviewers' renewed comments.

Reviewer 1

1. Comments and Suggestions for Authors
In general, it is a very interesting study. It is better for the authors to use more concise language and more intuitive graphs to show the amazing results, rather than too many numbers in the tables and a lot of descriptions of the tables.

The comments of both reviewers were similar in terms of changing the presentation of the obtained results. As suggested by reviewer no.2 the presentation of results from all sampling (T1, T2 and T3) was omitted. The authors reported the averages for the main effects (season, maintenance system, addition of polyunsaturated fatty acids), which contributed to increasing the transparency of the experiment results.

2. Quality of English: In the manuscript, some language editing is required to correct the grammar and syntax errors. Some expressions need to be more accurate and concise to avoid confusing readers.

Every effort has been made to improve the quality of the language and the legibility of the text.

3. Abstract: More information on methods and results should be provided here.

Following the reviewer's suggestions, information has been added, which should improve the readability of the abstract.

4. Statistical analysis: It was a three-factor study, and each factor with two levels. Therefore, the statistical analysis methods need to be explained in detail. I don’t think the sample size was big enough to use standard deviation, because the size of every single sample was just 8.

The presentation of results from all sampling (T1, T2 and T3) was omitted. The authors reported the averages (± sd) for the main effects (season, maintenance system, addition of polyunsaturated fatty acids). In accordance with these changes a new statistical test was carried out.  Information about the statistical test was added in section: 2.7. Statistical analysis

5. Feeding: The authors need to provide the formulation of the basal diet, as this is related to the performance of the rabbit fur.

The animals were fed in accordance with the feeding standards of reproductive rabbits during the period of sexual dormancy. The granulate was purchased from a local animal feed company. The granule formulation is the property of the company. The authors hope that the data on the concentration of nutrients will be sufficient information on the characteristics and quality of the feed used.

6. Fig. 1: Which group did the data come from in every stage?control (K) or experimental (D)? And the same question for Fig. 2.

Before starting linseed oil ethyl esters supplementation, the proportion of cover and down hair (%) and heat transfer coefficient (W / mK) of rabbit coat were assessed on control groups. This information was added to the figure description. The authors would like to thank you for catching this inaccuracy.

7. Table 3: What does the spiral symbol mean?

Spiral symbols, such as: â’¶, â’· shown differences between experiment studies, and symbol A, B shown differences between each samples.
Due to the change in the results presentation, it is no longer necessary to use duplicate determination of statistical differences.

8. 3.2. Histological analysis of hair: I think you can organize the pictures better for easy reading and understanding.

Following the reviewer's suggestions, the photos have been grouped into tables. The authors hope that thanks to this, the results of the histological analysis are easy reading and understanding for the reader.

The authors hope that the corrections made in line with the reviewers' suggestions have improved the readability and quality of the manuscript.

Yours faithfully,
Authors

Reviewer 2 Report

Introduction too long and often not in line with the purpose of the study

Materials and Methods

you must report the average temperature and humidity recorded during the experiment.

 fiber –14.82%; type of fiber, crude fiber or ADF ?

you have to report the statistical model.

Results

tables 3-4-5 are difficult to understand. you have to report the averages of the main effects (season, maintenance system, addition of polyunsaturated fatty acids).

The comparison between stages is also affected by the different ages of the animals.

The authors in the discussion never refer to the different ages of rabbits.

Author Response

Replies to reviewer - animals-1658374

The authors would like to thank the reviewers for all comments and suggestions which may improve manuscript quality. The authors made changes to the text in line with the reviewers' renewed comments.

Review 2

1. Comments and Suggestions for Authors
Introduction too long and often not in line with the purpose of the study

Following the reviewer's suggestions, the introduction was cut and the information not connected with the purpose of the study were omitted.

2. Materials and Methods
you must report the average temperature and humidity recorded during the experiment.

The information about average temperature and humidity were added.

3. fiber –14.82%; type of fiber, crude fiber or ADF ?

The concentration of nutrients, including crude fiber, was determined in the granulate. This information has been specified in the text.

4. you have to report the statistical model

AND

5. Results
tables 3-4-5 are difficult to understand. you have to report the averages of the main effects (season, maintenance system, addition of polyunsaturated fatty acids).
The comparison between stages is also affected by the different ages of the animals.
The authors in the discussion never refer to the different ages of rabbits.

The comments of both reviewers were similar in terms of changing the presentation of the obtained results. As suggested, the presentation of results from all sampling (T1, T2 and T3) was omitted. The authors reported the averages for the main effects (season, maintenance system, addition of polyunsaturated fatty acids), which contributed to increasing the transparency of the experiment results.
In accordance with these changes a new statistical test was carried out. Information about the statistical test was added in section: 2.7. Statistical analysis

The authors hope that the corrections made in line with the reviewers' suggestions have improved the readability and quality of the manuscript.

Yours faithfully,
Authors

Round 2

Reviewer 2 Report

The authors modified the manuscript as suggested by the reviewers. The manuscript has been improved, so it can be published in its current form

Author Response

The Authors would like to thank Reviewer for all coments which improve the value of manuscript.

Your faithfully,

Authors